# Rebuilding Stakeholder Confidence in Health-Relevant Big Data Applications: A Social Representations Perspective

**Anthony M. Maina [1,*] and Upasana G. Singh [2]**

[1] School of Computer Science and IT, Dedan Kimathi University of Technology, Private Bag, Nyeri 10143, Kenya
[2] School of Management, IT, and Governance, University of KwaZulu-Natal, Durban 4000, South Africa
* Correspondence: anthony.maina@dkut.ac.ke

**Abstract:** Big data applications are at the epicentre of recent breakthroughs in digital health. However, controversies over privacy, security, ethics, accountability, and data governance have tarnished stakeholder trust, leaving health-relevant big data projects under threat, delayed, or abandoned. Taking the notion of big data as social construction, this work explores the social representations of the big data concept from the perspective of stakeholders in Kenya's digital health environment. Through analysing the similarities and differences in the way health professionals and information technology (IT) practitioners comprehend the idea of big data, we draw strategic implications for restoring confidence in big data initiatives. Respondents associated big data with a multiplicity of concepts and were conflicted in how they represented big data's benefits and challenges. On this point, we argue that peculiarities and nuances in how diverse players view big data contribute to the erosion of trust and the need to revamp stakeholder engagement practices. Specifically, decision makers should complement generalised informational campaigns with targeted, differentiated messages designed to address data responsibility, access, control, security, or other issues relevant to a specialised but influential community.

**Keywords:** big data; big data technologies; digital health; health policy; social representations theory





## 1. Introduction

Big data is at the epicentre of recent developments in digital health systems [1–3]. With an expansive scope, it covers "biological, clinical, environmental, and lifestyle information" [4] about individuals or populations linked to their health and originates from sources as diverse as doctor's medical notes, medical images, laboratory test results, health insurance data, electronic patient records, biometric data, articles in medical journals, social media, to genomic databanks [5]. However, its availability to communities (for example, clinicians, data scientists, developers, and researchers) relies on established trust between various state and non-state actors [6,7].

Trust is seen as a prerequisite to successful data sharing projects and influences whether citizens and stakeholders will support a data sharing programme initiated by the government, health organisations, or research institutions [7–9]. However controversies over privacy, security, ethics, accountability, and data governance [10–12] have tarnished stakeholder trust, leaving health-relevant big data projects under threat, delayed, or abandoned [13–15]. Kenya's *Huduma Namba* (Swahili for "service number")—a project proposing to collect new citizens' and residents' personal information and link it with information held by other government agencies, such as public hospitals and government-supported health insurance—derailed because of legal hurdles, debates surrounding the collection of biometric data, and perceived lack of transparency in data management [16]. According to Morley et al. (2019), realising the benefits of large-scale data initiatives remains problematic because of a fundamental problem: "a deficit of trust" [17].

What factors influence trust in data-intensive, health-relevant applications? Privacy and security considerations dominate the arguments surrounding the lack of trust in

health-oriented, big data applications [11,18,19]. Other considerations include transparency, accountability, data ownership, data control, anonymisation, accuracy of scientific models, credible conduct of scientific research, and weak legislation [8,12,20]. In a study investigating enablers and impediments of trust in digital health systems, Adjekum et al. (2018) summarise three key facets: personal factors, such as *convenience*, *usefulness*, and *social demographics*, which drive individual decisions; technological factors, such as *privacy*, *interoperability*, and *customisable design*, which are technical attributes; and institutional factors, such as *improved communication* and *stakeholder engagement*, that describe strategies spearheaded by institutions, notably, government and health organisations [21].

This study considers the institutional factors influencing trust in health-relevant big data applications. We are interested in how government-led strategies could be enhanced to promote stakeholder trust in health-relevant big data applications. Using the social representations framework, we explore how a section of digital health communities in Kenya, comprising health and information technology (IT) practitioners, consider the big data concept. Social representations are the collective depictions or understandings of a reality common to a group of people [22,23]. The idea of "collective depictions" or "understandings" refers to attitudes, feelings, beliefs, symbols, perceptions, interpretations, values, and expectations that are common to people, constructed in interactions among community members and shared in the form of theories about our social world. Such representations are interpretations or reconstructions of a social reality rather than reflections of that reality.

The social representation lens is helpful for capturing and analysing "common sense" or "everyday" common knowledge and ideas about a social reality and, thereafter, drawing implications for policy, research, or practice [24]. In the social representation domain, distinct groups are neither knowledgeable nor ignorant of a phenomenon; rather, communities know (as a social construct) different things about it—in this case, big data, including its benefits, opportunities, challenges, and risks. Thus, understanding how big data is understood and perceived by a cross-section of stakeholders offers opportunities to illuminate the efficacy of initiatives promoting stakeholder trust and how the differences in big data perceptions underlie trust deadlocks.

Health practitioners are influential in the uptake of innovations within the health domain [25]. Along with IT experts, these communities anchor the digital health agenda. Significantly, these groups were key components in developing Kenya's national eHealth policy [26]. Two questions framed the inquiry: (1) How do health and IT professionals make sense of big data? (2) What are the similarities and differences in their perspectives? By analysing the emergent social representations of big data, we hope to uncover underlying beliefs and philosophies and draw strategies for boosting stakeholder confidence. Without trust, data-intensive health projects are impeded. Findings will be useful to decision makers and leaders in government and health-related organisations who wish to strengthen their digital strategies to support large-scale data innovations.

The next section discusses the general research approach. Thereafter, we explore the study's significant results and discuss strategic recommendations. The last section summarises the study's key findings, discusses limitations, and proposes future work.

## 2. General Empirical Approach

An empirical study was designed to draw experiences and immediate concerns of Kenyan stakeholders towards the notion of big data. It adopted an interpretivism stance and relied on a mixed method social representations methodology proposed by Jung et al. (2009) [24].

### 2.1. Data Collection and Analysis

Stakeholders were selected from two groups: health professionals (members of a national association of doctors) and information technology (IT) practitioners (members of one of the largest IT professional groups and staff of a large state corporation in the ICT

sector). The data were collected between March 2019 and January 2020 using the free word association technique and semi-structured interviews.

The free word association technique is common in social cognitive studies exploring individuals' thoughts, perceptions, and attitudes [27,28]. It opens the "conscious and unconscious part of an individual's mind, spontaneously expressing the first thoughts, images, and feelings that come to mind" [29]. Specifically, it works by presenting a stimulus word and asking a respondent to freely associate or correspond the word with ideas that come to mind, thereby, unlocking underlying "mental representations" [28]. For this study, respondents were asked to come up with three words or phrases they associate with the phrase "big data". This technique is quicker to administer than other approaches (for example, interviews or focus groups), and because digital data are captured, it has no requirement for transcription. Nevertheless, it yields less rich qualitative data than interviews or focus groups. Respondents were selected purposively, and a survey tool emailed or sent via WhatsApp address. The list also expanded via the "snowball" technique.

Semi-structured interviews flowed based on respondents' concerns but still covered the same topics: (1) How do they describe big data? (2) What are the benefits of big data in healthcare? (3) What are the challenges of using big data in health? (4) How can those challenges be overcome? The interviews lasted between 20 to 40 min each and were recorded and transcribed. We purposely selected interview respondents from the participants of the word association survey.

Data analysis employed the methodology used by Jung, Pawlowski, and Wiley-Paton (2009) [24] and comprised (1) coding to identify the key concepts of social representation using qualitative content analysis, (2) analysis of the structure of representation using analysis of similarity and core/periphery analysis, and (3) correspondence analysis to illustrate the perceptual space of social representations visually (Figure 1).

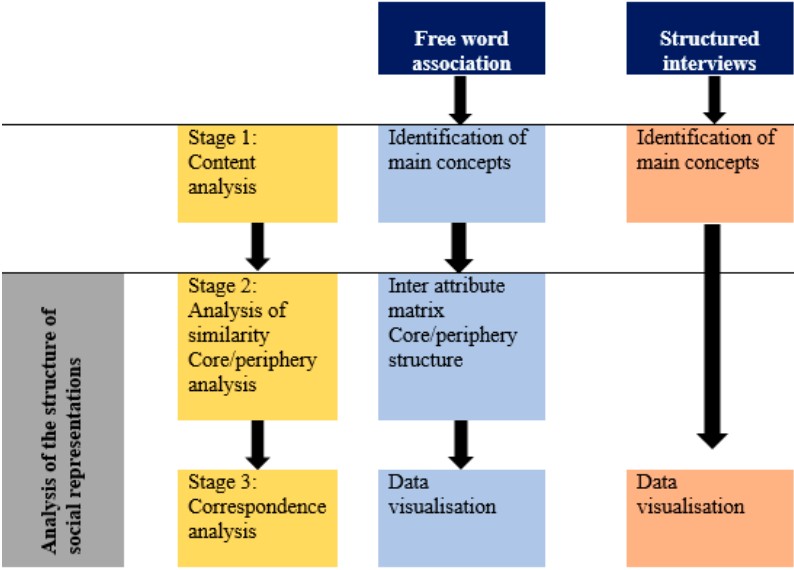

**Figure 1.** Social representations methodology.

## 2.2. Quality and Ethics

Two considerations guided the decision to adopt a variety of data collection and analysis techniques. First, in a multi-stage project, a mixed method design offers the opportunity to use varied approaches at different stages [30]. In this case, content analysis was foundational to introducing the quantitative analytical techniques: core/periphery analysis, similarity analysis, and correspondence analysis. Finally, method triangulation—the use of multiple data collection methods to study the same phenomenon [31]—compensates for the weakness of a single approach. The free word association technique, for example, provided for data collection from a higher number of respondents than would have been

possible if only interviews were employed. On the other hand, interviews were employed because they offer a rich data source compared to word association.

Because of different epistemological assumptions, the notion of reliability and validity in qualitative research tends to differ from the idea of these terms in quantitative studies [32]. To ensure reliability, the researchers counter-checked the interview transcriptions to eliminate errors and used a second coder to achieve coding consistency.

Research protocols were reviewed by the University of KwaZulu-Natal's Humanities and Social Sciences Research Ethics Committee (approval number HSS/2091/018D).

## 3. Findings

### 3.1. Word Association Survey

Out of the 105 valid responses, government-affiliated IT staff (40%, n = 42) had the highest representation, while private-sector doctors (12%, n = 13) had the lowest (Table 1).

**Table 1.** Sample Demographics.

|  | Government | Private Sector | Total |
|---|---|---|---|
| IT | 42 (40%) | 29 (28%) | 71 (68%) |
| Health | 21 (20%) | 13 (12%) | 34 (32%) |
|  | **IT** | **Health** |  |
| Female | 14 (13%) | 12 (11%) | 26 (25%) |
| Male | 57 (54%) | 22 (21%) | 79 (75%) |
| **Total respondents** |  |  | **105** |

#### 3.1.1. Big Data: Content Analysis

Using the open coding method, various themes were derived from the data. An independent reviewer re-coded the data using the codes identified during the initial coding. The investigators agreed on 136 codes out of the initial 150 codes (Cohen's Kappa = 0.906), representing high-level inter-rater reliability [33]. A final list of 130 codes was agreed upon and summarised through discussion and consensus into 17 topic themes (Table 2).

**Table 2.** Social Representations of Big Data.

| Topic Theme | Sample Codes | Topic Theme | Sample Codes |
|---|---|---|---|
| *T1 Size/Volume* | massive, large data sets, high volume, huge, colossal, many variables, tonnes of data | *T9 Security* | security, privacy, hacking, encryption, loss, no privacy |
| *T2 Data* | raw data, data, streaming data | *T10 Unstructured* | multimedia, unstructured data, social media, disorganisation |
| *T3 Technology* | information technology, ICT, blockchain, Hadoop, infrastructure, servers | *T11 Insight* | information, strategic, knowledge, evidence, patterns, trends, preferences |
| *T4 Applications* | Google, research, survey, policy, YouTube | *T12 Opportunities* | transformational, potential, value, gold rush, monetisation, research potential, business opportunities |
| *T5 Analytics* | integration, analysis, aggregation, processing, data mining, download, consolidation, predictive analytics | *T13 Time* *T14 Cost* | time, long time, different times expensive, money |
| *T6 Speed* | velocity, data stream, data explosion, fast | *T15 Cloud/Internet* | cloud, online, internet, connectivity |
| *T7 Complexity* | variety, complex, multiple sources, diversity | *T16 Smart/Artificial Intelligence* | smart logic, artificial intelligence, intelligence, learning, habits |
| *T8 Storage* | databases, data repositories, warehouse, data lake, hard drive | *T17 Fourth Industrial Revolution* | 4th industrial revolution |

#### 3.1.2. Big Data: Similarity Analysis and Core/Periphery Analysis

The next stage involved analysis of similarity and core/periphery analysis. An analysis of similarity assumes that the more closely topics are used together, the closer they are to a social-cognitive structure [34]. It was assessed using an inter-attribute similarity (IAS) matrix (Tables 3 and 4). Table cells contain a Jaccard's similarity coefficient that denotes the level of co-occurrence (proximity) for a given pair of attributes [35]. The sum of similarity was calculated as the row or column sum of this matrix; the higher the similarity total, the closer its association with other topics [24].

**Table 3.** Inter-attribute Matrix (IAS)-A.

|  | T1 | T2 | T3 | T4 | T5 | T6 | T7 | T8 |
|---|---|---|---|---|---|---|---|---|
| **T1** | 1.000 | 0.354 | 0.049 | 0.047 | 0.125 | 0.108 | 0.095 | 0.083 |
| **T2** | 0.354 | 1.000 | 0.000 | 0.000 | 0.255 | 0.027 | 0.075 | 0.114 |
| **T3** | 0.049 | 0.000 | 1.000 | 0.111 | 0.022 | 0.000 | 0.050 | 0.000 |
| **T4** | 0.047 | 0.000 | 0.111 | 1.000 | 0.043 | 0.000 | 0.000 | 0.036 |
| **T5** | 0.125 | 0.255 | 0.022 | 0.043 | 1.000 | 0.000 | 0.064 | 0.167 |
| **T6** | 0.108 | 0.027 | 0.000 | 0.000 | 0.000 | 1.000 | 0.056 | 0.042 |
| **T7** | 0.095 | 0.075 | 0.050 | 0.000 | 0.064 | 0.056 | 1.000 | 0.071 |
| **T8** | 0.083 | 0.114 | 0.000 | 0.036 | 0.167 | 0.042 | 0.071 | 1.000 |
| **T9** | 0.023 | 0.000 | 0.000 | 0.105 | 0.091 | 0.000 | 0.048 | 0.217 |
| **T10** | 0.108 | 0.056 | 0.000 | 0.000 | 0.047 | 0.000 | 0.056 | 0.042 |
| **T11** | 0.173 | 0.137 | 0.091 | 0.056 | 0.204 | 0.063 | 0.083 | 0.023 |
| **T12** | 0.024 | 0.026 | 0.063 | 0.118 | 0.022 | 0.000 | 0.000 | 0.000 |
| **T13** | 0.056 | 0.061 | 0.000 | 0.000 | 0.050 | 0.000 | 0.143 | 0.000 |
| **T14** | 0.000 | 0.000 | 0.000 | 0.000 | 0.000 | 0.125 | 0.000 | 0.000 |
| **T15** | 0.000 | 0.025 | 0.118 | 0.050 | 0.067 | 0.133 | 0.048 | 0.037 |
| **T16** | 0.024 | 0.054 | 0.063 | 0.056 | 0.122 | 0.000 | 0.000 | 0.000 |
| **T17** | 0.000 | 0.000 | 0.000 | 0.000 | 0.026 | 0.000 | 0.000 | 0.000 |
| **Sum of similarity** | **2.270** | **2.183** | **1.565** | **1.620** | **2.303** | **1.553** | **1.788** | **1.831** |

**Table 4.** Inter-attribute Matrix (IAS)-B.

|  | T9 | T10 | T11 | T12 | T13 | T14 | T15 | T16 | T17 |
|---|---|---|---|---|---|---|---|---|---|
| **T1** | 0.023 | 0.108 | 0.173 | 0.024 | 0.056 | 0.000 | 0.000 | 0.024 | 0.000 |
| **T2** | 0.000 | 0.056 | 0.137 | 0.026 | 0.061 | 0.000 | 0.025 | 0.054 | 0.000 |
| **T3** | 0.000 | 0.000 | 0.091 | 0.063 | 0.000 | 0.000 | 0.118 | 0.063 | 0.000 |
| **T4** | 0.105 | 0.000 | 0.056 | 0.118 | 0.000 | 0.000 | 0.050 | 0.056 | 0.000 |
| **T5** | 0.091 | 0.047 | 0.204 | 0.022 | 0.050 | 0.000 | 0.067 | 0.122 | 0.026 |
| **T6** | 0.000 | 0.000 | 0.063 | 0.000 | 0.000 | 0.125 | 0.133 | 0.000 | 0.000 |
| **T7** | 0.048 | 0.056 | 0.083 | 0.000 | 0.143 | 0.000 | 0.048 | 0.000 | 0.000 |
| **T8** | 0.217 | 0.042 | 0.023 | 0.000 | 0.000 | 0.000 | 0.037 | 0.000 | 0.000 |
| **T9** | 1.000 | 0.000 | 0.000 | 0.125 | 0.000 | 0.000 | 0.111 | 0.000 | 0.000 |
| **T10** | 0.000 | 1.000 | 0.000 | 0.071 | 0.000 | 0.000 | 0.000 | 0.000 | 0.000 |
| **T11** | 0.000 | 0.000 | 1.000 | 0.000 | 0.069 | 0.036 | 0.057 | 0.029 | 0.037 |
| **T12** | 0.125 | 0.071 | 0.000 | 1.000 | 0.000 | 0.000 | 0.000 | 0.067 | 0.000 |
| **T13** | 0.000 | 0.000 | 0.069 | 0.000 | 1.000 | 0.200 | 0.000 | 0.000 | 0.000 |
| **T14** | 0.000 | 0.000 | 0.036 | 0.000 | 0.200 | 1.000 | 0.091 | 0.000 | 0.000 |
| **T15** | 0.111 | 0.000 | 0.057 | 0.000 | 0.000 | 0.091 | 1.000 | 0.000 | 0.000 |
| **T16** | 0.000 | 0.000 | 0.029 | 0.067 | 0.000 | 0.000 | 0.000 | 1.000 | 0.000 |
| **T17** | 0.000 | 0.000 | 0.037 | 0.000 | 0.000 | 0.000 | 0.000 | 0.000 | 1.000 |
| **Sum of similarity** | **1.721** | **1.379** | **2.057** | **1.516** | **1.578** | **1.452** | **1.736** | **1.415** | **1.063** |

Next, we used core/periphery analysis to identify the core (main/central) and periphery (auxiliary) themes using the parameters of expressive value and associative value [36]. Expressive value measures salience or variable frequency—calculated by adding the frequency of topic themes in the data set [36].

The fundamental principle of associative value is that the core elements are associated with more elements compared to the peripheral components [36]. It is measured by the attributes of sum of similarity and coreness, as described previously by Jung et al. (2009). Table 5 shows the sum of similarity, salience, and coreness for each topic theme.

**Table 5.** Core/Periphery Structure of Big Data.

| | Topic Theme | Sum of Similarity | Salience (Weighted Frequency) | Coreness | Core/Periphery |
|---|---|---|---|---|---|
| **T5** | **Analytics** | **2.30** | **16.41** | **−0.412** | |
| **T1** | **Size** | **2.27** | **14.02** | **−0.422** | **CORE** |
| **T2** | **Data** | **2.18** | **11.68** | **−0.435** | |
| **T11** | **Insight** | **2.06** | **12.31** | **−0.341** | |
| **T8** | Storage | 1.83 | 8.17 | −0.273 | |
| **T7** | Complexity | 1.79 | 5.33 | −0.238 | |
| **T15** | Cloud/Internet | 1.74 | 4.42 | −0.172 | |
| **T9** | Security | 1.72 | 4.25 | −0.188 | |
| **T4** | Applications | 1.62 | 4.83 | −0.147 | |
| **T13** | Time | 1.58 | 0.96 | −0.164 | |
| **T3** | Technology | 1.57 | 3.92 | −0.140 | PERIPHERY |
| **T6** | Speed | 1.55 | 2.50 | −0.155 | |
| **T12** | Opportunities | 1.52 | 3.92 | −0.112 | |
| **T14** | Cost | 1.45 | 0.67 | −0.090 | |
| **T16** | Smart/AI | 1.41 | 3.87 | −0.133 | |
| **T10** | Unstructured | 1.38 | 3.25 | −0.137 | |
| **T17** | 4th Industrial revolution | 1.06 | 0.25 | −0.026 | |

### 3.1.3. Big Data: Interpretations of Social Representations

Four elements—analytics (T5), size (T1), data (T2), and insight (T11)—form the core elements of big data (Table 5). Core or central elements are assumed to have a higher recurrence rate than peripheral elements [13]. Analytics (T5) is the strongest theme and the most frequently cited. It was referenced using many concepts, among them "integration", "analysis", "aggregation", "processing", "data mining", "download", "consolidation", "extraction", and "predictive analytics". The second highest frequency theme was Size (T1), formed by some of the following concepts: "large data sets", "volumes of data", "huge", "tonnes of data", "colossal", and "many variables". We summarised the topic themes further as descriptive (for example, data, size, unstructured, speed, and time); risks (for example, security, complexity, and cost); infrastructure (for example, cloud/Internet, smart/AI, and technology); and benefits (for example, insight, opportunities, applications, and analytics).

### 3.2. Semi-Structured Interviews

Semi-structured interviews were conducted face-to-face (9) and via telephone (7). Interviewees—ten IT practitioners and six doctors—were purposively selected from the survey's 105 respondents. A majority (n = 11) had more than ten years work experience, while the rest had worked for 6 to 10 years in their discipline.

### 3.2.1. Big Data: Content Analysis

We used qualitative content analysis to identify the main topic themes. Based on open coding, 28 main concepts/topics were derived, representing the social cognitive structure of big data in healthcare. We organised the topic themes into four categories: definitions, benefits, challenges, and solutions (Figure 2).

Data analysis continued using correspondence analysis to aid the understanding of social representations. In correspondence analysis, data is graphically mapped to uncover "the nature of the associations among the variables", supporting the interpretation of social cognitive structures [37]. The perceptual map generated demonstrates the associations among the main themes and between health and IT professionals, the two main demographic variables.

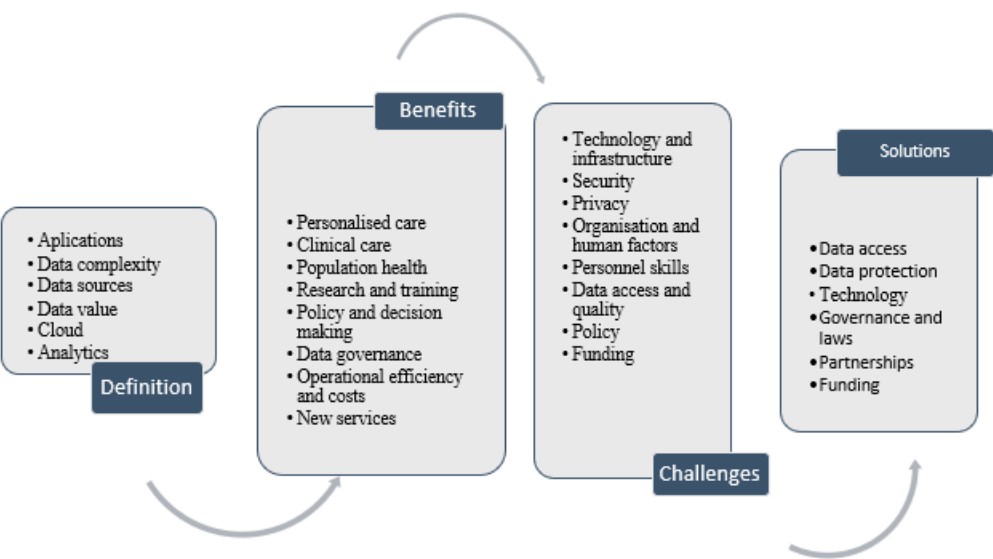

**Figure 2.** Social Representations of Big Data.

3.2.2. Correspondence Analysis: Defining Big Data

Big data was defined using six concepts: data value, applications, analytics, cloud, data complexity, and data sources (Figure 3). The two axes account for 61.3% of the data's inertia (variance). The concepts symbolise the nature of big data, including how it is captured, processed, and used.

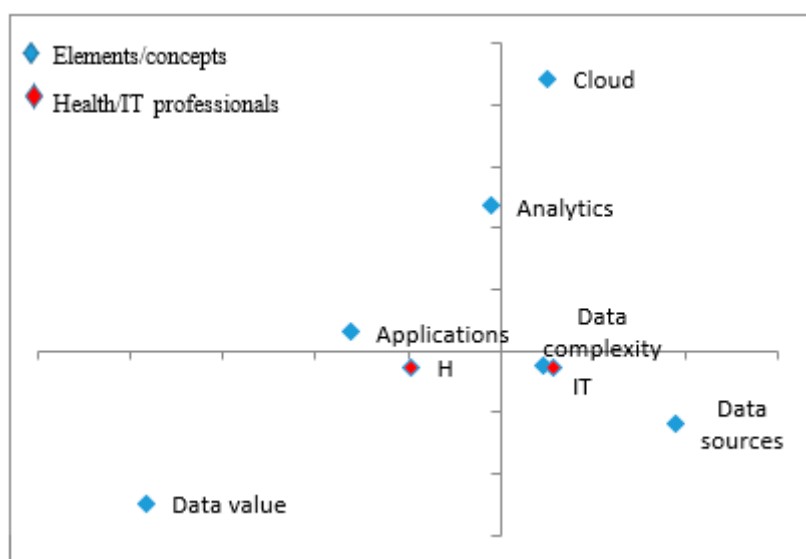

**Figure 3.** Defining Big Data.

For health professionals, big data is first and foremost seen based on its contribution to the clinical environment and health services and was associated with "Applications" and "Data value" concepts. A doctor expressed the following view:

*My lay understanding of big data is that it starts with collection of data, for example, temperature, blood pressure, and so on, from a single patient. Then you end up collecting such data from many patients ... Eventually, you accumulate information to help you answer questions like, 'What are the variations in body temperature for certain diseases?' Big data puts together the bigger picture of what is happening ... —Interviewee #13*

A health manager added:

*It's all about being able to collect data about service delivery . . . Big data analytics then looks at the data that has been collected over time and uses the same data to . . . improve health services.—Interviewee #16*

IT professionals associate big data with "Data complexity" and "Data sources." Their understanding is skewed towards big data's technical aspects, emphasising its unstructured format, massive volume, and real-time capture. An IT executive stated as follows:

*Traditionally data was structured. Data was structured until social media came, and we discovered that through social media everyone is a generator of information or data . . . So when we talk about big data it's about all these uncorrelated data which is everywhere, being generated by different people, being generated by devices, in an unstructured way.—Interviewee #7*

A software developer added:

*When you talk about big data you are talking about large amounts of data that have been collected over time and stored somewhere. And where is this data coming from? It's coming from transactions either due to human interaction or due to items that have been programmed.—Interviewee #9*

### 3.2.3. Correspondence Analysis: Benefits of Big Data Technologies

Health big data benefits are summarised into eight topics: "Personalised care", "Clinical care", "Population health", "Health research and education", "Policy and decision making", "Data governance", "Operational efficiency and costs", and "New services" (Figure 4). The two axes account for 52.8% of the data's inertia (variance).

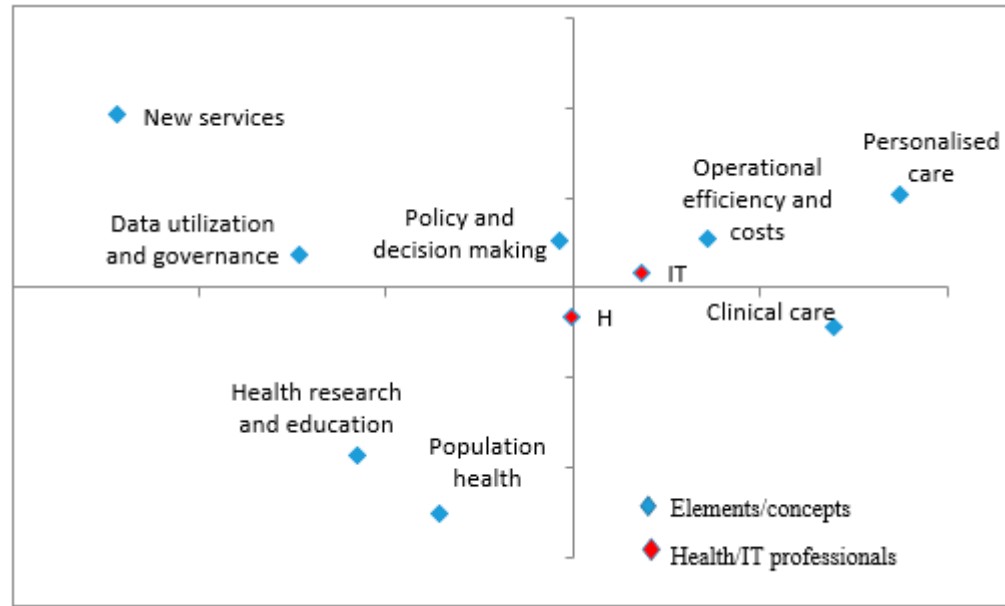

**Figure 4.** Benefits of Big Data.

There is a varied emphasis on the advantages of big data innovations: health communities favour health research and education and population health; IT practitioners support clinical care, operational efficiency and personalised care (Figure 4).

### 3.2.4. Correspondence Analysis: Challenges of Big Data Technologies

Big data barriers were visualised based on the following themes: "Technology and infrastructure", "Security", "Privacy", "Organisational and human factors", "Personnel skills", "Data access and quality", "Policy and regulation", and "Funding/Resources" (Figure 5). The two axes account for 56.5% of the data's inertia (variance).

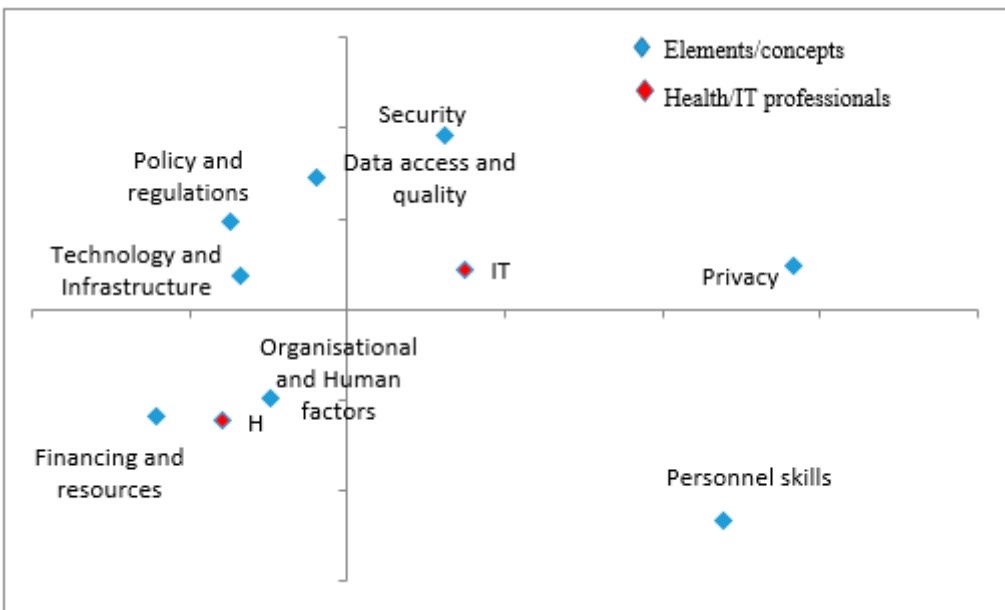

**Figure 5.** Challenges of Big Data.

The emerging relationships demonstrate that health respondents are linked most strongly with financing and resources and organisational and human factors. At the same time, IT participants show the strongest association with the challenges of privacy and personnel skills (Figure 5).

Health professionals singled out resource constraints as significant barriers to advancing big data technologies in healthcare. A physician argued:

*One of the challenges that we face when it comes to scaling up the use of these technologies is the issue of budget constraints. A majority of projects rely on donor funding, and therefore, these initiatives are not sustainable once the donor is unable to continue.— Interviewee #14*

Limited resourcing is aggravated by poor planning and other human factors, notably, unawareness of the benefits of novel technologies. A doctor highlighted the following:

*The main challenge is the lack of resources. But it is also about resource allocation, lack of prioritisation, and limited understanding about the benefits of these technologies. There are other things considered more basic such as drugs, so that when it comes to things such as the internet it is not considered as basic although they may play a role in making services better.—Interviewee #12*

The IT professionals' predominant issue is privacy. Additionally, IT workers asserted that the talent and skills necessary to harness the opportunities in health big data are lacking. An IT specialist observed the following:

*How many people are in technology and understand health informatics? Very few . . . and those few ones are stretched. County health records officers are trained to keep the records, not run analytics. Data may be exist . . . but unless they are trained on how to carry out data analytics, they can only wait for somebody else to do it.—Interviewee #8*

In addition to privacy, IT practitioners are strongly associated with another related issue: security. An IT manager described the following scenario:

*So let's say you visited the hospital and underwent some tests . . . results show you have heart problems and you are given a pacemaker to help your heart keep the rhythm. You perhaps only want your close family members know your health status. Let's say that that pacemaker is connected to the internet as part of the IoT so that it can update your doctor on your status, how it (pacemaker) is working, and its battery and so on . . .*

*if a bad guy happens to know that you have this pacemaker and it's connected to the internet, he could have the device hacked and try to control your life. This can turn out to be a life-and-death issue that you don't want to contemplate—it seems farfetched but possible.—Interviewee #7*

### 3.2.5. Correspondence Analysis: Solutions to Big Data Challenges

Six key topics were identified as possible solutions to big data problems in healthcare: "Data access", "Technology", "Governance and laws", "Partnerships and training" and "Funding and innovation". The two axes in Figure 6 account for 69.2% of the data's inertia (variance).

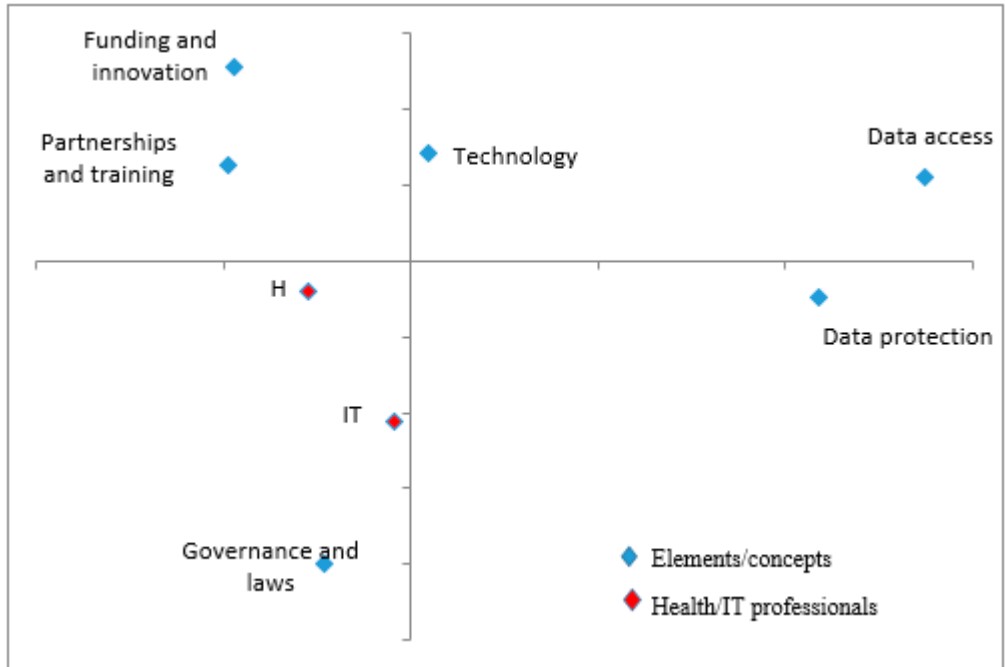

**Figure 6.** Solutions to Challenges of Big Data Technologies.

The outcomes from the qualitative analysis demonstrate that health practitioners are associated most strongly with partnerships and training and funding and innovation, while IT shows the strongest association with governance and laws (Figure 6). Health respondents highlighted solutions that correspond to the challenges they identified as the most dominant. Specifically, partnerships and training can tackle skills inadequacy and negative staff attitudes, whereas funding can address budget constraints. A physician remarked:

*Technology is still seen as a luxury . . . However, technology is still relevant. If the government allocated more resources in terms of internet and connectivity, that would be helpful. Other players too have responsibility. They can pick and strengthen areas that the government has not been able to address . . . —Interviewee #14*

Similar to health professionals, IT practitioners present possible solutions that align with their raised problems. For instance, governance and laws are relevant to strengthening privacy laws and can catalyse supporting data specialists' education and training. In particular, IT professionals emphasise the role of government in the establishment of an effective policy and regulatory environment, as noted in the following response:

*Traditionally government have existed for policy. So the government needs to provide the relevant policies and laws . . . Additionally, the government needs to also put in the right environment, for example, in terms of incentives so that the private sector to roll out such projects. The ripple effect would be massive gains for the populace.—Interviewee #12*

## 4. Discussion

Digital health systems are witnessing a transformation driven by the rise of large-scale data innovations, or so-called health big data applications. Notwithstanding, there is growing public and stakeholder mistrust over a myriad of personal, technological, and institutional factors [21]. Using the theoretical base of social representations, we investigated how a group of Kenyan stakeholders, comprising health and IT practitioners, comprehend the big data phenomena. Having an interest in institutional factors, we theorised that the notion of big data is a social construction and that beliefs and values surrounding it—though limited to the views of participating stakeholders—offer insights into why trust in big data projects is problematic while also identifying possible remedies.

First, the study observed that big data beliefs are anchored in understanding standard digital health data. In total, 17 concepts (for example, opportunities, applications, improvement, security, cost, disruption, wireless, and internet) formed the emergent social representations of big data. In the social representation domain, plurality and the non-specific nature of the concepts are evidence of the dominance of anchoring—the mechanism of coping with unfamiliarity [22]. The essence of anchoring is that when faced with a new concept or phenomenon, individuals often lack a representation to characterise it meaningfully and, thus, name or identify a new phenomenon based on what is already familiar. In conceptualising big data ideas, respondents were associating, or anchoring, big data with health data as used with electronic health records, mHealth and telemedicine. Nonetheless, representing big data innovations with popular health information technologies may misrepresent its potential benefits, opportunities, and challenges [2,24]. For instance, health big data transcends typical health system boundaries and, hence, faces more significant risks of privacy violations and security breaches [9].

Second, stakeholders are seemingly predisposed to big data's hype and opportunities. Big data was associated with ideas such as insight, speed, opportunities, smart/AI, and 4th Industrial revolution. For instance, insight was cited nearly three times more than security. A speculative bubble surrounds big data innovations because of the optimism around new technologies [38,39]. Some commentators suggest that technologies will eliminate uncertainties in healthcare, introduce ground-breaking hypotheses for new studies, and revolutionise clinical decision making [2,39]. Unfortunately, too much hype feeds unrealistic expectations, damaging trust, particularly when projects fail or under-deliver [40]. Moreover, technology hype may drive hasty innovation deployment, often without strict regard to privacy, security, or other regulatory concerns. As Newlands et al. (2020) succinctly assert, "rapid innovation and regulatory compliance . . . often make poor bedfellows" [41].

Finally, evidence shows that the perception of big data is influenced by interested parties' education and professional background. While not a particularly striking observation, it explains why the stakeholders were conflicted in how they made sense of big data technologies. As is typical in healthcare, multiple parties have wide-ranging viewpoints about emergent privacy and ethical concerns, including the nature and design of collaborations. Prior studies have also found evidence of conflicting stakeholder viewpoints, not the least on data management and governance [42,43]. Considering that stakeholders' confidence grows out of the anticipated benefits to the health system, mixed understandings are likely to undermine the general confidence.

The significance of trust in digital health, and especially, big data applications cannot be overemphasised. Stakeholder considerations have characterised the rise or fall of big data projects, from the recent roll out of COVID-19 contact tracing apps [15] to earlier initiatives such as the *Huduma Namba* in Kenya [44], and the *care.data* in the UK [13]. Directly or indirectly, it is hard to miss the common thread of lack of trust that faced these beleaguered systems. Specifically, the confounding issues appear domiciled under the category of trust elements identified by Adjekum et al. (2018) as "institutional factors"—most notably government-led strategies (for instance, stakeholder engagement and communication) promoting the uptake of advanced digital health systems. In his analysis of Kenya's *Huduma Namba* project, Mwaura (2019) observed that the stakeholders criticised

its lack of transparency in how the information was used and were apprehensive about the involvement of external entities, typically technology companies [44]. It was also apparent that the government's information machinery failed to properly support how the system would work and how the information would be shared among involved entities. Waning in stakeholder confidence, the *Huduma Namba* system faltered, and its implementation was eventually abandoned [45]. In the case of the UK's controversial *care.data* programme, critics cited an unclear agenda and bungled communication as some of the reasons that led to the project's downfall. The project "failed to earn the trust and confidence of patients, citizens, and health professionals" [9].

While lack of trust is acknowledged as an impediment to advancing large scale data projects [7,9], countermeasures mostly target privacy [11], security [20], and legislative reform. While making reference to the *Huduma Namba* project, Kimani (2019) asserts that the unique challenge low- and middle-income countries (LMIC) face in the adoption of data-intensive projects was that such innovations are "rarely preceded by the enactment of robust legal frameworks" [46]. Gopichandran et al. (2020), assessing the ethical and privacy issues affecting India's Aadhaar system, argue that "low and middle income countries must invest in developing strict legal regulations to protect data and avoid its exploitation for profit" [47]. Granted, there is a place for robust security mechanisms, more secure algorithms, and tighter legislative controls. However, it is also apparent that such measures and reforms—at least on their own—are insufficient to reverse the tide of wavering stakeholder confidence.

As seen in the study, stakeholders' big data perspectives are influenced by educational and professional backgrounds, yet this seems to have little bearing on implementation strategies. The government-led campaign on the *Huduma Namba* project relied on generalised messages to health professionals, IT experts, and other interested parties [48]. However, is this strategy realistic with many stakeholder groups, who have diverse ways of making sense of big data? Crucially, it is an approach that overlooks the impact of respondents' varied perspectives or views. A generalised communication strategy, in effect, assumes that the stakeholder community has a common meaning or similar understanding about technology. In a study investigating the threats posed by COVID-19 vaccine hesitancy among health workers and the general Israeli population, Dror et al. (2020) established that there were high levels of scepticism among medical staff not directly caring for infected patients [49]. The researchers proposed targeted advocacy to counter misinformation among groups at high risk of vaccine hesitancy. In concert, Vergara et al. (2021) argued the case for a "localised public education" as a response to the lack of trust in COVID-19 vaccines [50]. In this respect, complementing generalised informational campaigns with focused communication that addresses specialised groups' concerns or challenges could be the key to unlocking trust deadlocks.

Is introducing layered communication overburdening already expensive projects? Obviously, multipronged informational programmes will bear additional costs, but this must be weighed against the likelihood of failure when key parties reject the projects. In fact, cost of failure through inept communication might be higher. More importantly, the localised or targeted communication approach only applies to the key or influential stakeholders, who vary based on the nature of the project [51]. In the case of *Huduma Namba*, winning over health and IT professionals will be an essential step. The communication should address proposals on legal reforms, partnerships with technology companies, data collection, access to data, and confidentiality, among other areas of controversy. Additionally, advocacy to influential groups can clarify proposed projects' objectives, scope, and limitations and mitigate the risks of technology hype. Finally, while poor communication is not the only domain to address in resolving trust concerns, it is one of the most significant. In its analysis of the significance of communication in public education, the World Health Organisation (WHO) identifies six determinants of trust: competence, objectivity, fairness, consistency, sincerity, and faith [52]. A multilevel strategy, we argue, is best suited to support these

tenets of trust and establish a robust advocacy programme to advance health-relevant big data applications.

## 5. Conclusions

Repairing stakeholder trust will be critical to reinvigorating large-scale data applications in health. This work suggests why stakeholder confidence in data-intensive health-relevant projects is problematic: first, because health big data is associated with conventional digital health data; second, because the hype surrounding big data tends to raise impractical expectations; and finally, because the diversity of health domain actors leads to multiple perceptions of privacy, data security, and other ethical challenges. In view of various meanings attached to big data, findings question the validity of standardised messaging to popularise and promote health big data projects. The "one size fits all" communication approach, we argue, overlooks peculiarities in how diverse players perceive technology and is a likely culprit in the erosion of trust. While corroborating the significance of stakeholder trust in health data projects, this research calls for remodelling of the communication approaches by complementing generalised informational campaigns with targeted communication to critical communities such as health professionals and IT experts.

The study was not without limitations. First, the sample of respondents limited the extent to which generalisations could be made. Future work could incorporate social representations of other key groups (for example, patients) to broaden our understanding of trust considerations in digital health systems. Second, considering that the research was a cross-sectional study, the dynamic nature of social representations has implications for findings. Longitudinal studies may provide avenues to identify changes in understandings of big data over time. Finally, future studies could explore the broader scope of policy actions required to advance health big data in various jurisdictions. For example, though some countries have revised their data protection regulations, implications for data-intensive health innovations are unclear.

**Author Contributions:** Conceptualization, A.M.M. and U.G.S.; methodology, A.M.M.; software, A.M.M. and U.G.S.; validation, A.M.M. and U.G.S.; formal analysis, A.M.M.; investigation, A.M.M.; resources, A.M.M.; data curation, A.M.M.; writing—original draft preparation, A.M.M.; writing—review and editing, A.M.M. and U.G.S.; visualization, A.M.M.; supervision, U.G.S.; project administration, U.G.S.; funding acquisition, U.G.S. All authors have read and agreed to the published version of the manuscript.

**Funding:** This research received no external funding.

**Data Availability Statement:** Not applicable.

**Conflicts of Interest:** The authors declare no conflict of interest.

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
