# Peer review of "Rebuilding Stakeholder Confidence in Health-Relevant Big Data Applications: A Social Representations Perspective"

_information, doi:10.3390/info13090441_

Round 1

Reviewer 1 Report

This initiative explores the social representations of the big data phenomena from the perspective of stakeholders in Kenya’s digital health landscape. Such exploration is carried out by analyzing the similarities and differences in the way health professionals and information technology (IT) practitioners comprehend the notion of big data.

Although the presented study investigates the social representations of big data and draws implications for promoting stakeholder confidence in health-related data-driven innovations. The main contribution of this work is not clear. It is necessary to specify in an explicit way, the main contribution of this research.

In the introduction, the authors mentioned that two questions framed the inquiry: 1) How do health and information technology (IT) professionals make sense of big data? and, 2) What are the similarities and differences in perspectives their perspectives?. But on what basis do the authors consider these two questions as key to their research? It is necessary to justify.

A Related Work section is necessary, as well as to introduce a comparative table in order to know the advantages and drawbacks in comparison with other similar works. Also, authors should include related works with high-quality references, such as indexed journals.

In Data Collection and Analysis section. The authors mentioned that they carried out semi-structured interviews that included the following topics: 1) How do they describe big data?; 2) What are the benefits of big data in healthcare?, 3) What are the challenges of using big data in health?; 4) How can those challenges be overcome?.

 Similarly, these previous observations are necessary to justify, Why these types of questions? In the sense that questions are oriented to health professionals and IT specialists. Specifically, do you consider that healthcare professionals are aware of the use of Big Data in their institutions to answer such questions?

 The general empirical Approach section is incomplete.

 An empirical Approach is based on a Life Cycle that not is included in this section. Such Life Cycle must include: 1) Observation: this is necessary for proposing a hypothesis or research questions (the questions of paper); 2) Induction:  It is then carried out to form a general conclusion from the data gathered through observation; 3) Deduction: It is in order to deduce a conclusion out of their experiments; 4) Testing: It involves to return to empirical methods to put the hypothesis (or research questions) to the test. The authors can include a Case Study and, 5) Evaluation: It is an important piece of knowledge.

Hence, it is necessary to restructure Section 2. 2. General Empirical Approach and section 3 Findings.

 Although in the section Introduction, the future work section is mentioned, however, this information is not included in the paper.

Reviewer 2 Report

This is an interesting research to explore the concerns of public backlash to big data regarding to the worry of over privacy and ethical considerations. The authors interviewed 105 health professionals and information technologists. They concluded “we argue that peculiarities and nuances in how diverse players view big data contribute to the erosion of trust and the need to revamp stakeholder engagement practices. In particular, decision makers should complement generalised informational campaigns with targeted, differentiated messages designed to address data responsibility, access, control, security, or other issues relevant to a specialised but influential community.”

This is a well-written manuscript; however, I suggested the authors to clarify:

1.     How did they invite the participants? How about the response rates for health professionals and information technologists? Is it representative?

2.     Depending on the methods they used, the limitations of this study should be discussed.

Reviewer 3 Report

The paper presents a study of social representation of Big Data focusing on two categories of people: IT and Health professionals.

The  discussion is consistent and the paper is easy to read, however it appear as a case study and the scientific contribute it is not clear.

The results might be to biased by the selection and number of interviewed people (The authors report it as possible limit into the conclusions, however I suggest a deeper discussion and some proposal to overcome the issue) 

Reviewer 4 Report

Rebuilding stakeholder confidence in data-driven health initiatives: A social representations perspective

In this study, authors investigate stakeholders’ perspectives (Doctors and IT specialists) on big data in health care using two methods: Word Association Survey and interview and report the identified themes.

I think more information regarding social representations perspective is needed.

You used the phrase “data-driven” in title but your study focused on big data.

Please correct this sentence: “differences in perspectives their perspectives”

There is little information about Digital health in Kenya in Introduction to comprehend the context of the study.

Only doctor? And IT? What is their experience in digital health? Did you have any criteria for selection of participants?

Rigor (for qualitative study) should be betted explained in the Methods.

Please pay attention to numbering the tables.

Analysis methods are mentioned in the results, not in Methods section.

In qualitative study, researchers considered Data saturation to stop interviews. How did you select 16 interviewees? Criteria for interview participants?

In Fig 3-6, what are the x and y axes and values? Please describe these figures.

Related studies and gaps and your contribution in the introduction should be better introduced.  

In discussion, Kenya digital health (big data) status should more discussed.  

You mentioned that “A multilevel strategy, we argue, is best suited to support these tenets of trust and establish a robust advocacy programme to support data-driven applications, particularly in healthcare.” Although your study was completely in health sector, why did you conclude about other data-driven applications?

Round 2

Reviewer 1 Report

The authors have addressed all my comments. The content of this paper has been improved significantly. I can recommend this paper for publication.

Author Response

Thank you for your feedback. We appreciate your assistance in improving the quality of this manuscript. 

Reviewer 4 Report

I would like to thank author for considering most of my previous comments in the revised manuscript. I have no new comment for the manuscript.  

Author Response

(The authors gave the same response as above.)
